# Association of Socioeconomic Factors and Physical Activity with Health-Related Quality of Life in Italian Middle School Children: An Exploratory Cross-Sectional Study

**DOI:** 10.3390/healthcare11142092

**Published:** 2023-07-22

**Authors:** Francesco Sanmarchi, Lawrence M. Scheier, Laura Dallolio, Matteo Ricci, Giulia Longo, Andrea Ceciliani, Alice Masini

**Affiliations:** 1Department of Biomedical and Neuromotor Sciences, University of Bologna, 40126 Bologna, Italy; francesco.sanmarchi@studio.unibo.it (F.S.); matteo.ricci18@studio.unibo.it (M.R.); giulia.longo9@studio.unibo.it (G.L.); 2LARS Research Institute, Inc., Sun City, AZ 85351, USA; scheier@larsri.org; 3Prevention Strategies, Greensboro, NC 27410, USA; 4Department of Life Quality Studies, University of Bologna, Campus of Rimini, 47921 Rimini, Italy; andrea.ceciliani@unibo.it; 5Department of Translational Medicine, University of Eastern Piedmont (UNIUPO), Via Solaroli, 17, 28100 Novara, Italy; alice.masini@uniupo.it

**Keywords:** public health, quality of life, adolescents, physical activity

## Abstract

Health-related quality of life (HRQoL) provides a broad assessment of an individual’s well-being and can serve as a good prognosticator of life’s outcomes later for children and adolescents. Understanding the factors associated with HRQoL is crucial for promoting better health and life satisfaction. This study investigated the cross-sectional association of socioeconomic status, cardio fitness, and physical activity levels with HRQoL in 224 Italian early adolescents attending secondary school in the Emilia-Romagna region located in Northern Italy. In a multivariate path regression model, younger students and females reported a higher quality of life (β = −0.139, *p* = 0.015, 95% CI: −0.254–−0.023 and β = 0.142, *p* = 0.019, 95% CI: 0.011–0.273, respectively). Having both parents employed and having a higher familiar educational status were also associated with a higher quality of life (β = 0.142, *p* = 0.013, 95% CI 0.027–0.257 and β = 0.133, *p* = 0.017, 95% CI 0.022–0.244, respectively). Greater engagement in routine physical activity levels from moderate to vigorous intensity was positively associated with quality of life (β = 0.429, *p* < 0.001, 95% CI: 0.304–0.554). Endurance (speed) was positively associated with quality of life (β = 0.221, *p* = 0.001, 95% CI: 0.087–0.355), and students with longer times on the shuttle run reported less quality of life (β = −0.207, *p* = 0.002, 95% CI: −0.337–−0.077). These relations remained intact even when controlling for socioeconomic factors. The current findings reinforce the importance of promoting regular physical activity among younger children and also addressing socioeconomic status disparities to improve children’s well-being. Future studies may want to consider expanding the array of measures used to assess physical activity and include additional measures assessing nutrition, cultural factors, and family functioning, all of which can influence a child’s willingness to engage in physical activity and their well-being. The emphasis on fitness and physical activity and their contribution to a child’s well-being should be the prime focus for stakeholders who work in the education, public health, and health policy sectors.

## 1. Introduction 

The construct of health-related quality of life (HRQoL) represents a subjective interpretation of an individual’s well-being. As it is generally used, HRQoL encompasses a person’s health status, as well as non-medical aspects of functioning, including emotional, social, and cognitive functioning [1]. In general, most health professionals consider HRQoL as a valid and useful barometer of how happy a person is and how satisfied with life they may be [2]. Since its wider adoption in the clinical literature in the 1960s, the concept of HRQoL has proven useful in assessing satisfaction with treatment and disease remediation for a wide range of medical conditions. The idea behind HRQoL is that a person’s health is not merely the absence of disease but is more holistic and includes their physical, mental, and social well-being [2]. It is this fitness of mind and body that gives HRQoL a unique place in the various literatures attending to well-being, happiness, and life satisfaction [2].

The World Health Organization (WHO) Quality of Life Group set the tone for how HRQoL is conceptualized, defining quality of life as “the individual’s perception of their position in life in the context of the culture and value systems in which they live and in relation to their goals, expectations, standards and concerns” [1]. Accordingly, HRQoL is both an “ideal” and also a means to assess a person’s state of satisfaction, whether prior to illness, during treatment, or after recovery [3]. From a measurement standpoint, HRQoL assesses important aspects of health that are subjective and not detected by traditional physiological and clinical assessments. In this respect, HRQoL is regarded as a multidimensional construct [4] extending beyond a person’s reflection on their physical condition to include an appraisal of their emotional and psychosocial functioning in addition to their health and well-being [5,6].

Studies of HRQoL in children and adolescents have produced a wide array of findings [7,8,9]. Much of this work has been disease-specific, focusing on conditions like diabetes, asthma, cancer, obesity, learning disabilities, and mental health conditions [10,11,12,13,14]. These studies have greatly contributed to our understanding of the impact of such conditions on children’s well-being and satisfaction with life. Parallel to these studies has been a concerted effort to develop reliable measures of HRQoL in younger populations, with the aim of generating population-based data to inform policy and decision-making [15].

Although much of the early work using the WHO epidemiological perspective emphasized health in relationship to functional status and disability, both conceptual models of HRQoL and assessment strategies now incorporate both individual and environmental factors that influence well-being [16]. These latter refinements incorporate a more “ecological” perspective consistent with biopsychosocial metatheories of human development [17]. This broad and more inclusive conceptualization of HRQoL has led many to consider it an organizing construct that incorporates the qualities and conditions of an individual’s life, their subjective satisfaction, expectations, and values that guide their life [18].

From childhood onward, HRQoL takes on special meaning. During the early adolescent years, in particular, a growing child experiences rapid physical, cognitive, emotional, and social development. These maturational changes, along with powerful socialization factors, can influence a person’s growth and development and contribute to their well-being, including setting the tone for later life adjustments. Thus, assessing HRQoL can provide rich and meaningful insight into the mind and world of young children, particularly with respect to different aspects of their psychosocial functioning.

The World Health Organization (WHO) defines physical activity (PA) as any bodily movement produced by skeletal muscles that requires energy expenditure [19]. One consistent finding from previous research has been the positive association between PA and HRQoL in children and adolescents [20]. This most likely results from the function of PA, because it can improve a person’s outlook on life, leading to subjective reports of more favorable well-being. Several studies have shown that regular engagement in PA is associated with better physical, mental, and social health outcomes [21,22,23]. Additionally, higher levels of PA have been linked to improved self-esteem and decreased symptoms of anxiety and depression in both children and adolescents [24]. A positive relationship between PA and HRQoL is particularly relevant given consistent findings that young children today are becoming increasingly more inactive and sedentary than ever before [25]. Thus, promoting regular PA may be an effective strategy to improve HRQoL in children and adolescents.

Socioeconomic status (SES) and related demographic factors may also affect both a child’s engagement in PA, as well as influence their HRQoL [26,27]. This relationship arises because HRQoL reflects the status of a person, usually in the form of social comparison. In making these comparisons, usually upward in nature, individuals, including children, gauge how well they are doing juxtaposed against some standard of living. As a result, SES-related measures such as family income, parental education, family structure, and social support (as an indicator of social capital) may affect whether a child is happy, content, and balanced and whether they feel they have sufficient resources to obtain a high quality of life [28,29,30].

To illustrate, studies show that children and adolescents from lower SES backgrounds who experience some form of material deprivation have poorer HRQoL than their peers from higher SES backgrounds [31]. This follows the assumption that a social class gradient contributes to different HRQoL in children, as it does with adult health [32]. In support of this contention, parental education has been found to be positively associated with HRQoL, with higher levels of education being linked to better HRQoL in children and adolescents [33]. In addition to education, family material affluence may contribute to HRQoL [34]. Families with more social capital and financial resources can afford to engage their children in sport clubs that reinforce physical activities. More affluent families have greater accessibility to parks, the relative safety of neighborhoods, and available resources to support club activities for children. Conversely, families that face chronic adversity and who live in under-resourced neighborhoods cannot access the social and material resources that support organized sports and expose their children to health-engendering activities. As an extension of this premise, social support, including family and peer support, has been identified as an important factor in promoting HRQoL in younger populations [7]. The latter connection may reflect the advantages of social skills garnered in the presence of family and friends, higher self-concept, and social networks that buffer children from adversity and stressful life events. When surrounded by supportive family and friends, children tend to be optimistic and have a better outlook on life.

### Research Question and Purpose of the Study 

The “Active Breaks in Secondary School Children: The BRAVE Study” is a quasi-experimental study to implement and evaluate the efficacy of a PA intervention conducted in a secondary school setting. The study commenced in 2022 and was managed by teachers and peer educators. The PA sessions were conducted as mini “active breaks” during classroom time and coordinated to reinforce much of the learning materials (i.e., counting jumping jacks was tied to solving math problems).

The present study examines several determinants of HRQoL, focusing specifically on SES and healthy lifestyle factors (i.e., PA and cardio fitness (CF) performance) among Italian middle school students participating in the Active Breaks intervention. To accomplish this, participants were administered the Physical Activity Questionnaire for Older Children (PAQ-A) [35] and the KIDSCREEN-27 questionnaire [36]. Both instruments are very cost-effective, epidemiological assessments that have been validated with similar age populations. To the best of our knowledge, this is the first study to explore the combined effects of SES and PA on HRQoL in Italian middle school students. As such, the study is poised to fill an important gap in the literature and provide novel insights into whether there is a social gradient to HRQoL and whether PA and performance measures contribute uniquely to HRQoL in a younger age population. Ultimately, the results of this investigation can inform the development of targeted interventions aimed at promoting healthy development and well-being in younger children during a critical stage of development when children are more prone to engage in PA and which can favorably shape the remainder of their lives.

## 2. Materials and Methods

### 2.1. Study Design and Participant

The BRAVE study involved students attending a secondary school located in Valsamoggia (Bologna), part of the Emilia Romagna region of Northeastern Italy. The Bioethics Committee of the University of Bologna approved the “BRAVE” project on 18 March 2022 (Protocol n. 63053). The study was conducted in accordance with the Declaration of Helsinki. The research team obtained written informed consent from the parents of the participating children.

Participants were recruited from the first to third grades (corresponding to ages 12 to 14). The inclusion criteria stipulated that students were free of any health issues or physical disabilities that might interfere with or impact their PA performance. This was meant to avoid any health issues or physical disabilities that might significantly interfere with or impact students’ PA performance. These included significant cardiovascular, respiratory, or musculoskeletal disorders; neurological conditions; and severe visual or hearing impairments. These exclusion criteria were meant to ensure the safety of the participants and to avoid potential confounding that might influence the study outcomes. The study adhered to the STROBE guidelines [37].

### 2.2. Instruments

Data from the students were collected in January 2023 using a self-reporting questionnaire administered by the teachers of each class assessing their SES, along with basic demographic information and general PA. Separately, using a standardized protocol including an extensive training session, physical education teachers collected CF performance data.

### 2.3. Main Outcome Measure: Health-Related Quality of Life

An Italian version of the KIDSCREEN-27 questionnaire [38] was used to assess the students’ HRQoL. The KIDSCREEN-27 is a psychometrically validated and widely used epidemiological instrument appropriate for children and adolescents between 8 and 18 years of age [36]. It consists of 27 self-reporting items that assess their perceived health status and quality of life broken down into five domains: physical well-being (5 items: MacDonald’s ω = 0.80 for reliability) [39], psychological well-being (7 items: ω = 0.89), parent relations and autonomy (7 items: ω = 0.88), social support and peers (4 items: ω = 0.81), and school environment (4 items: ω = 0.72). The instrument has been tested in relatively large samples, demonstrated good scale reliability, has both construct and criterion validity in various cultures, and has been translated into several languages, rendering it a valuable tool for both clinical and population-based research purposes in the field of pediatric health [36].

The physical well-being domain assesses energy and vitality (e.g., “Did you feel full of energy?”), physical activity (e.g., “Were you able to run as fast as you wanted?”), and general health (e.g., “How do you rate your health?”). The psychological well-being domain evaluates psychological functioning, including self-esteem (e.g., “Were you happy being the way you are?”), emotional well-being (e.g., “Did you feel lonely?”), and body image (e.g., “Did you feel happy with the way you look?”). The parent relations and autonomy domain assess different facets of the child’s relationship with their parents, including communication (e.g., “did your parents have enough time for you?”), support (“Did you feel understood by your parents?”), and perceived parental control (“Did your parents let you make your own decisions?”).

The social support and peer domain assesses the child’s relationships with their peers (“Did you feel supported by your friends?”) and social environment (e.g., “Did you have fun with your friends?”). The school environment domain assesses the school climate (e.g., “Did you feel happy at school?”), teacher support (e.g., “Did you have good relationships with your teachers?”), and subjective academic performance (e.g., “Were you satisfied with your grades?”). Each item on the KIDSCREEN-27 questionnaire is scored on a five-point Likert scale ranging from 1 (“never”) to 5 (“always”), with scores for each domain calculated separately to provide an overall assessment of the child’s HRQoL (min 27 (low quality of life); max 135 (high quality of life)).

### 2.4. Physical Activity Questionnaire for Older Children

The Italian version of the Physical Activity Questionnaire for Older Children (PAQ-A) [40,41] was used to assess the children’s PA levels. The PAQ-A consists of nine items probing PA levels over the past 7-day timeframe. Children are instructed there are no right or wrong answers and to answer the questions honestly. The nine questions assess participation in physical activities that cause sweating or heavy breathing, daily activity duration, sedentary behavior, active commuting, fitness improvement efforts, sports proficiency, physical education class participation, non-school related screen time, and extracurricular activities. Responses are rated on a 5-point Likert scale ranging from 1 (“never”) to 5 (“always”). This instrument provides a final composite activity score by averaging the scores for the 9 items. The total score of PAQ-A ranges from 0 to 5 points. The PAQ-A has been shown to be valid and reliable with both children and adolescent samples [42].

### 2.5. Cardio-Fitness Performance Measurements

Cardio-fitness performance measures were collected using the 6-min Cooper test (assessed in meters) [43,44], a timed shuttle run test [45], and standing long jump (measured in centimeters) [46,47]. Cardio-fitness performance and PA are closely related, and both provide key indicators of health outcomes [48,49,50]. The 6-min Cooper test was conducted on a 28 × 15 m track, where participants were instructed to run as far as possible within 6 min (i.e., endurance). The shuttle run 4 × 10 test (4 10 SRT) [45] measures speed, agility, and coordination while the subject runs between two lines drawn on the floor 10 m apart to pick up small blocks. Each student ran as fast as possible from the starting line to the other line and returned to the starting line, crossing each line with both feet every time. This test was performed twice. A stopwatch was used to determine the time based on when the student crossed the end line with one foot. The standing long jump (SLJ) had each student stand behind the take-off line and jump forward as far as possible (i.e., core power). The distance was measured from the take-off line to the point where the back of the heel nearest to the take-off line landed on the mat or non-slippery floor. The test was repeated twice, and the fastest time was retained. These CF tests have been shown to be a reliable means to assess physical fitness in children [44,46,47].

### 2.6. Socioeconomic Variables

The socioeconomic variables included the child’s nationality and the parents’ citizenship, educational level, and employment level. The child’s nationality was coded as either native or foreign-born, with the latter category encompassing children born outside of Italy. Parents’ citizenship was operationalized in the same manner as their child’s. Parents’ educational level was measured using the highest degree obtained by either parent, ranging from no high school diploma (reference category) to university degree or higher degrees. This variable was recoded to high school or less vs. university or graduate training. Parents’ employment level was assessed by categorizing parents as either unemployed or employed (i.e., 1 = both parents were employed; 0 = at least one parent was not employed).

### 2.7. Covariates

In addition to the SES measures, several other covariates were used in the analysis to control for potential confounds that may also have contributed to HRQoL. These included the child’s sex and age (difference between the day the survey was administered and each participant’s date of birth), International Obesity Task Force (IOTF) category, and family living status. The IOTF category was used as a measure of childhood weight status, with categories including age-specific estimates of thinness, normal weight, and overweight/obese. Living with both parents was coded as a dichotomous variable, indicating whether the child lived with both parents or not.

### 2.8. Data Analysis

Participants’ characteristics and responses were summarized using the mean and standard deviation and absolute and relative frequencies, as appropriate. The distributional properties of the dependent variables were assessed graphically using density graphs and tested for normality with the Kolmogórov–Smirnov test. The Student’s *t*-test was used to compare the means of independent groups, Wilcoxon signed-rank test to compare ordinal variables in dependent groups, and chi-square test to compare dichotomous variables in independent groups. The strength of association between the CF, PA, and KIDSCREEN 27 subscales was examined using the Pearson’s product-moment correlation coefficient.

A structural path model was used to examine the relationships between the SES measures, CF, PA, and HRQoL. We tested these relations using four integrated model building steps:

(1) *Confirmatory Factor Analysis* (CFA): A confirmatory measurement model was used to examine the dimensionality of HRQoL using the five KIDSCREEN-27 subscale domains. A single latent factor (HRQoL) was hypothesized to cause the association among the five observed subscale domains. The model fit was based on the Comparative Fit Index (CFI > 0.95), Tucker-Lewis Index (TLI), Root Mean Square Error of Approximation (RMSEA ≤ 0.08), Standardized Root Mean Squared Residual (SRMR ≤ 0.05), χ^2^/df (<5.0), and magnitude of the standardized factor loadings [51].

(2) *Covariate Only Model*: Next, a model was tested positing the relations of the covariates alone (child’s sex and age, nationality, family living status, and socioeconomic variables) with the latent factor of HRQoL. Examining the covariates in the absence of the CF and PA measures helps to identify any potential suppressor relations and also determine which SES measures are uniquely associated with HRQoL. The model fit was based on the same indices as the CFA model, and the nonsignificant measures were removed.

(3) CF *and PA Model*: A third model posited relations between the observed CF performance measures (6-min Cooper test, shuttle run test, and standing long jump); PA levels (PAQ-A); and HRQoL. The same model fit indices were applied, and we examined the increment in the model R^2^ (variance accounted for in HRQoL) with the addition of the second set of measures, controlling for the presence of the covariates.

(4) *Combined Model*: We then combined the results of steps 1–3 and tested a path model positing relations between the SES, covariates, CF, PA, and HRQoL. This model included only measures that were significant from the prior steps. Additionally, we evaluated the R^2^ statistic to determine whether the additional measures improved the overall variance accounted for in HRQoL compared to the prior models. The statistical significance level to reject the null hypothesis was set as *p* < 0.05. The combined model included associations among the fitness measures but not the covariates. All data management and descriptive analyses were conducted using R version 4.2.2 (R Project for Statistical Computing) [52], and the correlation and path modeling used Mplus statistical software V 8.4 [53].

## 3. Results

### 3.1. Study Population

Table 1 summarizes the main characteristics of the 370 early adolescents that comprised the school-based sample. There were slightly more males (*n* = 188; 51%) than females (*n* = 182; 49%). The mean age of the sample was 12.76 ± 0.94 years (range 10.89 to 15.34). A total of 54 (24%) participants were categorized as overweight/obese, and the majority lived with both parents (*n* = 194; 87%). The majority of the sample (*n* = 186; 83%) had Italian citizenship. As for parental educational level, 69 (31%) mothers had a university degree. The same level of higher education was reported by 51 (23%) fathers. Overall, 146 (39.5%) of the participants lacked data for the CF performance tests, PA measures, or the self-reporting HRQoL measures. As a result, they had over 80% of their data missing. We used listwise deletion, as any type of model-based imputation would not recover accurate standard errors or produce efficient parameter estimates with such large amounts of missing data.

### 3.2. Correlation Analysis

Bivariate relations between the main variables of interest are presented in Table 2.

The five subscales of HRQoL were moderately related, and all of the correlations were significant (r_avg_. = 0.451). The largest of these relations was between Psychological Well-being and Parent Relations and Autonomy (r = 0.536), while the smallest in magnitude was between Social Support and Peers and School Environment (r = 0.256). As expected, routine engagement in PA (PAQ-A) was moderately related to Physical Well-being (r = 0.579), with a much smaller relationship with School Environment (r = 0.075). The CF performance measures were all substantially related to each other, with longer agility times (shuttle run) inversely related to the long jump (r = −0.727) and endurance (r = 0.496), and endurance was inversely related to agility (r = −0.569). Also as expected, being able to jump and demonstrate agility and endurance were all moderately related to routine engagement in PA. Among the five subscales of HRQoL, Physical Well-being had the largest association with the CF tests (r_avg._ = 0.374) and the PAQ-A score (r = 0.579), respectively. The remaining associations with the HRQoL scales and CF were much smaller in magnitude (r_avg._ = 0.100, r_avg._ = 0.104, and r_avg._ = 0.127 for the long jump, shuttle run, and Cooper endurance test, respectively).

The point-biserial correlations indicated that females reported less physical (r = −0.13, *p* < 0.10) and psychological well-being (r = −0.20, *p* < 0.01), less autonomy and favorable family relations (r = −0.14, *p* < 0.10), less positive peer relations (r = −0.04, *p* = 0.795), and a better school environment and learning context (r = 0.18, *p* < 0.05). They also reported shorter 6-min run times in the Cooper test (r = −0.33, *p* < 0.001), faster shuttle run results (r = 0.24, *p* < 0.01), and less participation in routine PA (r = −0.17, *p* < 0.05). Although not significant, females reported longer distances in the standing long jump (r = 0.10, *p* = 0.361).

Figure 1 shows the results of the CFA model testing the dimensional structure of HRQoL using KIDSCREEN-27. The fit of the model was adequate according to several benchmark criteria: χ^2^(5, N = 224) = 5.84, *p* < 0.001, χ^2^/df = 1.17, CFI = 0.996, TLI = 0.992, SRMR = 0.02, and RMSEA = 0.03 (95% CI: 0.000–0.100). The CFI indicates the amount of covariation in the data accounted for by the hypothesized model (compared to a baseline independence model that posits the manifest variables are uncorrelated), with larger values > 0.95 indicating a good model. Both the SRMR and RMSEA [54] indicate “badness of fit” (smaller values are better)—in the former case, comparing the hypothesized model to an unrestricted mean and variance covariance structure (i.e., mean square error of the estimated correlations), and the latter case based on a discrepancy statistic that considers the sample and model-implied covariance matrices. Monte Carlo simulations have shown these model fit indices perform well with small samples and small degrees of freedom [55,56]. As depicted, all of the factor loadings for the measured variables were sizable in magnitude (λ_avg._ = 0.603) and significant (*p* < 0.001). The largest loading was for Parent Relations and Autonomy (λ = 0.726, SE = 0.050), and the smallest loading was for Social Support and Peer Relations (λ = 0.530, SE = 0.060).

### 3.3. Path Model

With a well hypothesized and reliable latent factor of HRQoL, we next tested a path model using the Mplus program. Table 3 contains the structural path coefficients for the univariate, multivariate, and fully adjusted models.

The model with only SES measures and demographic covariates fit well: χ^2^(33) = 60.733, *p* < 0.001, χ^2^/df = 1.84, CFI = 0.892, TLI = 0.853, SRMR = 0.054, and RMSEA = 0.061 (95% CI: 0.036–0.085) and accounted for 17.2% of the variance in HRQoL. The benchmark model fit indices indicated several model refinements could be made that might improve the overall fit. However, Monte Carlo simulations with relatively small samples (<500) have shown that additional refinements obtained with specification searches are likely to be sample-specific and not accurately reproduce the population model [57]. Male students reported higher HRQoL (β = −0.204, *p =* 0.005, 95% CI: −0.346–0.062), as did those living with both parents (β = 0.164, *p =* 0.023, 95% CI: 0.023–0.305) and having both parents employed (β = 0.235, *p =* 0.002, 95% CI: 0.087–0.384). The lower portion of the table in the first column shows the univariate results for the CF performance and PA measures: χ^2^(21) = 98.527, *p* < 0.001, CFI = 0.765, TLI = 0.664, SRMR = 0.085, and RMSEA = 0.128 (95% CI: 0.103–0.154). This model accounted for 39.3% of the variance in HRQoL, essentially doubling the information we know about HRQoL. Being routinely engaged in PA was associated with a better HRQoL (β = 0.418, *p* < 0.001, 95% CI: 0.257–0.579), as was endurance (speed: β = 0.184, *p =* 0.001, 95% CI: 0.033–0.336). Agility was negatively related to HRQoL (more time to cross the endline means slower agility: β = −0.186, *p =* 0.048, 95% CI: −0.371–0.001).

We then combined the two models. joining the covariates with the CF performance and PA measures. The multivariate model showed an adequate fit: χ^2^(33) = 133.646, *p* < 0.001, χ^2^/df = 4.05, CFI = 0.733, TLI = 0.635, SRMR = 0.078, and RMSEA = 0.115 (95% CI: 0.095–0.136), although, by all accounts, this model could also be improved (R^2^ = 49.6%). In the multivariate model, younger students reported higher HRQoL (β = −0.139, *p =* 0.020, 95% CI: −0.254–−0.023). In contrast to the univariate model, females reported higher HRQoL (β = 0.142, *p =* 0.017, 95% CI: 0.011–0.273). Students with both parents employed reported a higher HRQoL (β = 0.145, *p =* 0.036, 95% CI: 0.031–0.259), as did students from homes with a higher educational status (β = 0.135, *p =* 0.039, 95% CI: 0.023–0.248). The associations of the CF performance and PA measures with HRQoL remained consistent in the multivariate model. That is, a greater engagement in routine PA was positively associated with HRQoL (β = 0.429, *p* < 0.001, 95% CI: 0.304–0.554). Endurance (speed around the track) was positively associated with HRQoL (β = 0.221, *p =* 0.018, 95% CI: 0.087–0.355), and students who took longer on the shuttle run (slower agility) reported less HRQoL (β = −0.207, *p =* 0.044, 95% CI: −0.337–−0.077).

The final model added associations among the CF performance and PA measures to control for their moderate association: χ^2^(45) = 214.342, *p* < 0.001, χ^2^/df = 4.76, CFI = 0.545, TLI = 0.545, SRMR = 0.120, and RMSEA = 0.130 (95% CI: 0.112–0.147). The far right-hand side of Table 3 shows that the unique associations between the SES, CF performance, and PA measures with HRQoL did not appreciably change with the addition of these associations. The overall R^2^ for the model was 51.2%. Students who took longer on the shuttle run had less speed and endurance (Cooper test: r = −0.468, *p* < 0.001) and likewise engaged in less routine PA (r = −0.366, *p* < 0.001). Students who engaged in more routine PA were faster during the Cooper test (r = 0.345, *p* < 0.001).

We also ran a model that replaced age with the IOTF score (under/normal weight vs. overweight/obese). Running a separate model was necessary, because the IOTF score is age-specific, and including both measures would muddy, if not confound, the interpretation. The initial model fit well, and several measures were removed for non-significance. A final trimmed model was adequate: χ^2^(33) = 110.15, *p* < 0.001, χ^2^/df = 3.34, CFI = 0.779, TLI = 0.699, SRMR = 0.078, and RMSEA = 0.102 (95% CI: 0.081–0.124). The significant parameters included parents’ employment status (β = 0.126, *p =* 0.026, 95% CI: 0.007–0.245), parents’ educational status (β = 0.118, *p =* 0.017, 95% CI: 0.007–0.229), the shuttle test (β = −0.132, *p* = 0.002, 95% CI: −0.260–−0.004), Cooper endurance test (β = 0.151, *p* = 0.001, 95% CI: 0.024–0.278), and general PA involvement (β = 0.446, *p* < 0.001, 95% CI: 0.329–0.563). The IOTF score was not significant (β = −0.134, *p =* 0.13, 95% CI: −0.246–−0.022). In this model, citizenship of the parents was marginally significant (β = −0.108, *p =* 0.06, 95% CI: −0.226–0.009). The model accounted for 49.8% of the variance in the HRQoL measures.

## 4. Discussion 

Subjectively assessing one’s HRQoL captures myriad functions in a person’s life. This can involve evaluating how one is functioning psychologically, physically, socially, and more generally, whether one feels content and good about oneself. This type of personal subjective assessment can have a heuristic value for adults, as well as children, albeit the nature of this introspective focus may vary considerably based on age. Understanding the factors that are associated with HRQoL in children is particularly important, as their satisfaction with different domains of functioning can be a reliable prognosticator of life’s later outcomes. Moreover, the ability to identify the factors associated with HRQoL also creates opportunities to develop a wide range of interventions to promote better health and life satisfaction, with implications for both morbidity and mortality. To address this concern, and in an exploratory manner, we assessed several factors believed to be associated with HRQoL in a sample of Italian children attending secondary school, a period in life that corresponds to early adolescence. This is a crucial period of development when children begin to form lifelong impressions of their health and well-being. It is also a time when they remain malleable with regards to their health orientations, open to suggestions about exercise, nutrition, and the benefits of healthy living. The measures assessed in the study included the children’s CF performance, assessing their agility, strength, and endurance, and a self-reporting measure assessing their participation in PA. We also included objective measures of their SES status, assessing the parents’ education; citizenship; employment; and the child’s sex, age, nationality, and living situation. A measure of body mass (IOTF) was modeled separately given some confounding with age. The inclusion of traditional measures of the SES is intended to capture the “social gradient” of health, which suggests that inadequate material wealth and personal resources and a lack of social capital can hinder a person’s ability to satisfy certain needs and diminish their well-being.

Consistent with previous findings, the results of the CFA model reinforced that HRQoL is multidimensional [58]. The loadings on the five subscale domains were balanced and moderate in size, suggesting that each subscale contributed relatively equally to defining HRQoL. This suggests that, for this age group, HRQoL best represents a mixture of multiple contexts of a child’s life, including their vitality, energy, general health, emotional state of mind, home atmosphere, family relations and sense of parental control, peer social relations, how they feel about school, whether it is a supportive place, and a personal appraisal of their academic performance. These five facets provide a glimpse of a child’s overall functioning—in particular, their subjective interpretation of well-being and satisfaction with life. The CFA measurement model provides “error-free” estimates of the factor loadings and reinforces that HRQoL can be conceptualized as a reliable single dimension. The same model also reinforces that KIDSCREEN-27 can be administered in an abbreviated form to assess HRQoL in young children while still capturing the full essence of what makes them satisfied with different facets of their life.

The sequence of the path models showed that, when examined in isolation, certain SES and demographic factors provide important pieces of information regarding HRQoL. These are not arbitrary covariates; rather, they provide insight into the lifestyles of these children and account for meaningful variations in HRQoL. When the full set of SES and demographic measures were modeled, sex, living situation, and parent’s employment status were significantly associated with HRQoL. Females reported higher HRQoL, as did children living with both parents and in homes where both parents were employed. In general, males reported significantly higher levels of physical and psychological well-being and satisfaction with family relations and parental control. One explanation for these differences may involve a “socialization” effect (i.e., reporting differences between boys and girls) that captures differences in how girls and boys perceive their state of mind, emotional well-being, and family relations. Girls in this age group have the added experience of menarche that can tip the balance and create an aura of emotional friction in the home or lead them to report less satisfaction with life.

Parents’ education and employment, both objective measures of social status, were consistently related to HRQoL and might represent a social gradient of health, with higher income and more affluent families reporting less disease and sickness overall [59]. Education has also been shown to be an important component of a child’s overall health and well-being [30]. In homes with more educated parents, there is likely greater health literacy, leading to more informed decisions about a family’s healthcare [60]. Furthermore, a higher educational status often correlates with increased SES resources, such as material wealth, social capital, and better access to healthcare. Education and nutrition are also closely intertwined, leading to healthier food choices, with less dependence on fatty foods or foods high in sugar content [61]. The consistent relations between several objective SES measures and HRQoL reinforce that, even among children [59] and youths, there is a social gradient for health, with more resources associated with a better quality of life [36].

The next model built on the prior model and added the CF and PA measures. In this combined model, the standing long jump measure was not significant. The loss of predictive efficiency may happen with multiple measures of cardio fitness, because there is an empirical overlap in what they assess. In other words, the 6-min Cooper and shuttle tests capture strength, endurance, and agility, leaving no room for an independent contribution from a measure of power (jumping far indicates a lower core strength). Notably, a self-reporting measure of PA was, by far, the most efficient predictor of HRQoL. The PAQ-A score consists of nine items that provide a comprehensive view of the child’s routine physical activities based on their 7-day recall. Overall, more active children reported higher HRQoL. This finding reinforces the importance of regular PA for secondary school students and the influence that engaging in routine PA has on the quality of their life [62]. Regular PA has known benefits on physical health, cognitive functioning, and social skills [63], contributing to enhanced well-being.

Replacing age with the IOTF score (indicating normal vs. obese weight) provided some interesting findings. Both variables cannot be modeled simultaneously, given the IOTF score is age-specific. Parents’ work and educational status were both significant, as in the previous models. When we decomposed the five subscales and examined the bivariate associations, only physical well-being was significantly different, with heavier children reporting less well-being (r = −0.206, *p* < 0.05).

The results of this study align with numerous findings in the literature, which posit that routine engagement in PA can lead to a number of favorable outcomes in children. Physical activity is associated with less sedentary behavior, optimizes physiological functioning, and bolsters cognitive performance [64,65]. The significance of these findings is amplified when considering the long-term implications of PA habits established during childhood and adolescence. By instilling the importance of regular PA at an early age, parents and schools can promote a trajectory of health-related choices, subsequently reducing the risk of chronic diseases and bolstering a child’s sense of well-being throughout their life course [66]. Thus, it is imperative for stakeholders across the educational, public health, and policy sectors to prioritize and allocate resources towards programs that facilitate access to, and engagement in, PA for this age group.

The current study is part of an overall research agenda to promote PA in young children through the introduction of “Active Breaks” during classroom time. The basic emphasis on active breaks is to find ways to weave in small doses of PA throughout the school day without interrupting normal curricular instruction. This concept has emerged as a promising approach to bolstering PA in both children and adolescents and enhancing their health outcomes [67,68,69]. An accumulation of evidence now shows that active breaks present a valuable intervention strategy to reduce sedentary behavior and improve cognitive functioning and can promote better academic performance [70,71,72,73]. Furthermore, Active Breaks have demonstrated favorable effects on both mood and social behavior and may serve as a catalyst for promoting positive attitudes toward PA among younger populations. It is worth noting that active breaks can be readily integrated into the existing school infrastructure, thereby rendering this type of program both feasible and cost-effective as a means to encourage PA and improve health outcomes among children and adolescents [74].

### Limitations

There are several limitations to this study worth noting. First, the study was seriously underpowered, which could undermine detecting significant effects in the path model. This is especially important with path models where Type II error rates will increase for model fit indices with decreases in sample size (albeit the 224 students in the current study exceeds the lower limit testing in simulation studies) [75]. Clearly, a larger sample is required to robustly test the path model and ensure precision in the parameter estimates, including obtaining accurate standard errors. A post hoc Monte Carlo simulation to determine the adequate sample size reinforced that a high percentage of the parameter values and SEs were well within their respective confidence intervals, with minimal bias. Despite the exploratory nature of this study, we can still learn a great deal about a wide range of SES, demographic, CF performance, and PA measures that contribute to HRQoL in young children. Second, the cross-sectional design and the lack of temporal relations precludes making causal statements about factors that contribute to HRQoL in young children. Longitudinal studies with appropriate statistical controls (which we are conducting currently) are needed to infer causality from observational data. There may also have been a modicum of selection bias given the reliance on a convenience sample, which can affect the generalizability of these findings. The sample, sourced entirely from a single province within Emilia-Romagna, may not be representative of the entire region or population of Italy. This geographical limitation may restrict the external validity of the findings. There was the limited triangulation of data with corollary reports other than the performance tests and self-reporting HRQoL. All of the SES information was subjective based on the children’s reports. All of these factors suggest that future studies should replicate the current findings, using more diverse and larger samples encompassing multiple provinces or regions, with the additional corroboration of key markers from external sources.

In addition to these concerns, there are many factors we did not model that may contribute to HRQoL. For one thing, the three CF performance measures represent only a subset of possible strength and agility measures. There are many different types of field tests that can be used to gauge physical fitness. The assessment of additional activities, particularly those more representative of daily life or sport-specific activities, could provide a more comprehensive understanding of the relationship between PA and HRQoL. Furthermore, the reliance on a single trained individual for data collection may introduce potential observer measurement bias, suggesting that future studies should consider using multiple trained PE teachers or standardized protocols to minimize such bias. Moreover, additional measures of culture, family functioning, diet, and nutrition may also account for variations in children’s well-being and their HRQoL. Larger families have to share limited resources, including the parents’ time, disposable income, and their social network, all of which can strain family relations. What a person eats, their activity habits, parental support, involvement in sport clubs, and amount of time they spend engaging in leisure or pleasurable activities can all contribute to HRQoL. Likewise, other SES-related factors (i.e., household income, affluence, and social capital) can make a material contribution to a child’s well-being. The omission of relevant variables was perhaps best indicated by the less-than-optimal fit of the different path models. Although a relatively large amount of variance was accounted for in HRQoL, the overall fit of the models indicated that variance remained unaccounted for by the precise model specification. Future studies may want to revisit this model and include additional conceptually important measures.

## 5. Conclusions

The current study contains several strengths, including the use of rigorous statistical methods, reliance on psychometrically refined and well-validated instruments, and inclusion of a diverse set of measures linked to HRQoL. The study findings indicate a pronounced impact of SES-related factors, regular PA, and CF performance on a child’s self-reported HRQoL, highlighting a deep connection between these important correlates and the overall well-being of children. Moreover, the findings suggest a distinct health gradient linked to the education and employment status of parents, with children of better-educated and employed parents reporting higher HRQoL. The insight obtained from this study can serve as a foundation for developing interventions targeted at improving socioeconomic disparities to achieve more equitable health outcomes for children across various demographic backgrounds.

Furthermore, the current study encourages the inclusion of a broader range of measures in future studies of HRQoL, including, but not limited to, cultural influences, nutrition, and family characteristics. We believe that longitudinal studies investigating causal relationships between these important drivers of HRQoL, along with indicators of material wealth (i.e., household income data), will provide a more comprehensive understanding of the numerous factors influencing children’s well-being. Additionally, the current study underscores the need for future research to evaluate the impact of targeted PA interventions and explore potential mediators and moderators of these relationships. The insights from such studies could be pivotal for stakeholders in the education, public health, and health policy sectors to formulate strategies that promote access to, and engagement in, PA, thereby fostering healthy choices in middle school children. Taken as a whole, the current study provides a clarion call for a comprehensive, multifactorial approach to fostering a healthier, more resilient generation of youths better equipped to navigate the challenges of adolescence, young adulthood, and beyond.

The study findings also have implications for national and international stakeholders who work in the education, public health, and health policy sectors. By prioritizing and allocating resources towards programs that facilitate access to, and engagement in, PA for middle school children, we can foster a trajectory of healthy choices in children, ultimately reducing the risk of chronic diseases and improving their well-being throughout their life course. Furthermore, addressing socioeconomic disparities may contribute to more equitable health outcomes for children. A reasonable goal would be to achieve this with children from different national or cultural backgrounds. Together, these efforts can pave the way for a healthier, more resilient generation of youths better equipped to navigate the challenges of adolescence, young adulthood, and even beyond to adulthood.

## Figures and Tables

**Figure 1 healthcare-11-02092-f001:**
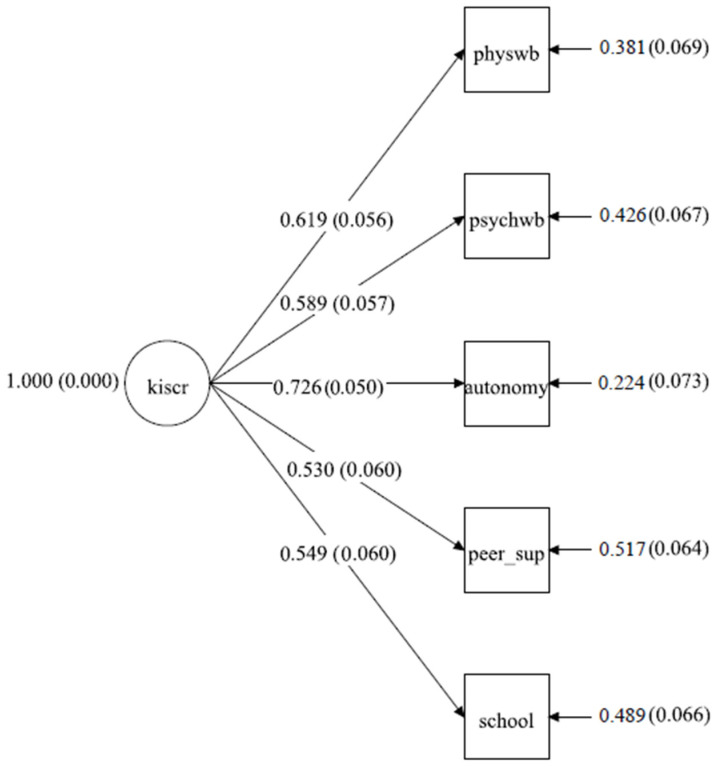
Results of the CFA analysis. Note: physwb = Physical Well-Being; psychwb = Psychological Well-Being; autonomy = Parents and Autonomy; peer_sup = Social Support and Peers; school = School Environment. The large circle denotes a latent factor, and the squares denote observed measures. The factor loadings are standardized. The numbers in parentheses for the factor loadings are standard errors and, for error terms, are squared residuals (net of prediction).

**Table 1 healthcare-11-02092-t001:** Population characteristics.

Population Characteristics	Overall, n= 370	Females, n = 182	Males, n = 188	*p*-Value ^3^
**Age (years, mean ± SD)**	12.76 ± 0.94	12.89 ± 0.91	12.63 ± 0.95	0.009
**IOTF category**				0.725
Underweight/Normal weight	170 (76%)	98 (77%)	72 (74%)	
Overweight/Obese	54 (24%)	29 (23%)	25 (26%)	
Missing	146	55	91	
**Live with both parents**				0.512
No	28 (13%)	18 (14%)	10 (10%)	
Yes	194 (87%)	108 (86%)	86 (90%)	
Missing	146	56	92	
**Nationality**				0.07
Italian	186 (83%)	111 (87%)	75 (77%)	
Other	38 (17%)	16 (13%)	22 (23%)	
Missing	146	55	91	
**Mother citizenship**				0.181
Italian	179 (81%)	106 (84%)	73 (76%)	
Other	43 (19%)	20 (16%)	23 (24%)	
Missing	146	56	92	
**Father citizenship**				0.7
Italian	193 (87%)	111 (88%)	82 (85%)	
Other	29 (13%)	15 (12%)	14 (15%)	
Missing	146	56	92	
**Parents citizenship**				0.228
Both Italians	173 (78%)	103 (82%)	70 (73%)	
One Italian-One other	26 (12%)	11 (8.7%)	15 (16%)	
Both other	23 (10%)	12 (9.5%)	11 (11%)	
Missing	146	56	92	
**Parents citizenship dichotomic**				0.159
Both Italians	173 (78%)	103 (82%)	70 (73%)	
Other	49 (22%)	23 (18%)	26 (27%)	
Missing	146	56	92	
**Mother educational level**				0.651
Middle school or lower	44 (20%)	26 (21%)	18 (19%)	
High school	109 (49%)	64 (51%)	45 (47%)	
University degree or higher	69 (31%)	36 (29%)	33 (34%)	
Missing	146	56	92	
**Father educational level**				0.442
Middle school or lower	60 (27%)	35 (28%)	25 (26%)	
High school	111 (50%)	66 (52%)	45 (47%)	
University degree or higher	51 (23%)	25 (20%)	26 (27%)	
Missing	146	56	92	
**Parents highest educational level**				0.651
Middle school or lower	39 (18%)	23 (18%)	16 (17%)	
High school	114 (51%)	67 (53%)	47 (49%)	
University degree or higher	69 (31%)	36 (29%)	33 (34%)	
Missing	148	56	92	
**Mother employed**				>0.999
Yes	34 (15%)	19 (15%)	15 (16%)	
No	188 (85%)	107 (85%)	81 (84%)	
Missing	146	56	92	
**Mother employment status**				0.105
Unemployed	34 (15%)	19 (15%)	15 (16%)	
Part time	70 (32%)	47 (37%)	23 (24%)	
Full time	117 (53%)	60 (48%)	57 (60%)	
Missing	146	56	93	
**Father employed**				>0.999
Yes	5 (2.3%)	3 (2.4%)	2 (2.1%)	
No	217 (98%)	123 (98%)	94 (98%)	
Missing	148	56	92	
**Family working status**				0.981
One employed or less	38 (17%)	21 (17%)	17 (18%)	
Both employed	184 (83%)	105 (83%)	79 (82%)	
Missing	148	56	92	
**Physical Well-Being (5) ^1^**	18.51 ± 3.38	18.02 ± 3.46	19.16 ± 3.19	0.013
Missing	146	56	92	
**Psychological Well-Being (7)**	27.66 ± 4.25	26.65 ± 4.47	28.94 ± 3.58	<0.001
Missing	146	61	93	
**Parents and Autonomy (7)**	28.09 ± 4.59	27.33 ± 4.92	29.07 ± 3.93	0.005
Missing	146	56	92	
**Social Support and Peers (4)**	16.73 ± 2.59	16.64 ± 2.58	16.85 ± 2.60	0.548
Missing	146	56	92	
**School Environment (4)**	14.27 ± 2.30	14.34 ± 2.20	14.19 ± 2.42	0.622
Missing	146	56	92	
**KIDSCREEN 27 total score**	105.79 ± 12.20	103.77 ± 12.38	108.37 ± 11.53	0.006
Missing	146	61	93	
**Long jump (cm) ^2^**	154.22 ± 33.08	144.13 ± 30.69	164.18 ± 32.43	<0.001
Missing	67	42	46	
**Shuttle run (s) ^2^**	12.13 ± 1.54	12.51 ± 1.38	11.77 ± 1.59	<0.001
Missing	96	49	47	
**6 min Cooper test (m) ^2^**	1807.28 ± 378.68	1645.75 ± 313.79	1977.64 ± 367.35	<0.001
Missing	105	47	60	
**PAQ-A score**	2.25 ± 0.62	2.08 ± 0.58	2.48 ± 0.58	<0.001
Missing	146	56	92	

Note: ^1^ Numbers in parentheses adjacent to scale name indicate the number of items in scale. ^2^ These variables are transformed in the path model by dividing them by a constant. ^3^ Comparisons by sex with dichotomous variables examined using Wilcoxon signed-rank test, with three or more levels by the chi-square test of independence, and continuous measures using the Students’ *t*-test.

**Table 2 healthcare-11-02092-t002:** Correlations between the IVs and DV (all 5 subscales).

Variable	Physical WB	Psych WB	Parents and Autonomy	Social Support and Peers	School Env.	KIDSCR. 27	PAQ-A Score	Long Jump	Shuttle Run	Cooper Test	Age
**Physical WB**	-	0.481	0.435	0.362	0.358	0.714	0.579	0.352	−0.408	0.361	−0.116
**Psychological WB**		-	0.536	0.349	0.453	0.824	0.141	0.181	−0.095	0.169	−0.210
**Parents and Autonomy**			-	0.422	0.375	0.805	0.206	0.073	−0.162	0.112	−0.070
**Social Support and Peers**				-	0.256	0.595	0.189	0.119	−0.073	0.146	−0.074
**School Environment**					-	0.606	0.075	−0.029	−0.092	0.082	−0.163
**KIDSCREEN 27**						-	0.313	0.192	−0.200	0.230	−0.158
**PAQ-A score**							-	0.263	−0.349	0.339	−0.092
**Long jump**								-	−0.727	0.496	0.144
**Shuttle run**									-	−0.569	−0.062
**Cooper test**										-	−0.022
**Age**											-

**Table 3 healthcare-11-02092-t003:** Standardized coefficients from the path model.

Structural Path Coefficients									
	Univariate Model ^1^			Multivariate Model			Fully Adjusted Model ^3^		
Measure	*b*	*p*-Value	95% CI	*b*	*p*-Value	95% CI	*b*	*p*-Value	95% CI
**Covariates**									
Age	−0.136	0.064	−0.280–0.008	−0.139	0.020	−0.254–−0.023	−0.136	0.019	−0.250–−0.023
Nationality	−0.058	0.477	−0.220–0.103	– ^2^	-		-	-	
Sex	−0.204	0.005	−0.346–−0.062	0.142	0.017	0.011–0.273	0.140	0.015	0.017–0.263
Living situation	0.164	0.023	0.023–0.305	– ^2^	-		-	-	
Parents’ employment status	0.235	0.002	0.087–0.384	0.145	0.036	0.031–0.259	0.142	0.013	0.027–0.257
Parents’ educational status	0.032	0.658	−0.111–0.176	0.135	0.039	0.023–0.248	0.133	0.017	0.022–0.244
Parents’ citizenry (immigration status)	−0.043	0.616	−0.212–0.126	– ^2^	-	-	-	-	
**Cardio-Fitness and Physical Activity Measures**									
PAQ-C (PA)	0.418	<0.001	0.257–0.579	0.429	<0.001	0.304–0.554	0.421	<0.001	0.300–0.542
Cooper Test (speed)	0.184	0.017	0.033–0.336	0.221	0.018	0.087–0.355	0.218	0.001	0.089–0.346
Long jump	0.011	0.919	−0.199–0.221	– ^2^	-		– ^2^	-	
Shuttle run (agility)	−0.186	0.048	−0.371–−0.001	−0.207	0.044	−0.337–−0.077	−0.203	0.002	−0.329–−0.077

Note: ^1^ Model parameters are for the fully saturated model not trimmed for nonsignificant effects. ^2^ Variables with no statistical information in the multivariate or fully adjusted models were trimmed for non-significance. ^3^ This model includes correlations among continuous independent measures. *β* = standardized coefficients; CI = 95% confidence interval. A model that replaced age with the IOTF produced trivial differences in the parameter estimates.

## Data Availability

Data are available upon request to the contact author.

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
