# Peer review of "Association of Socioeconomic Factors and Physical Activity with Health-Related Quality of Life in Italian Middle School Children: An Exploratory Cross-Sectional Study"

_healthcare, 2023, doi:10.3390/healthcare11142092_

Round 1
Reviewer 1 Report
First of all, congratulations for the work done, then I will mention a number of changes and recommendations in order to obtain clearer and more accurate information.
- Comments on the abstract:
Avoid the use of abbreviations in the abstract, and the if you use them, define the abbreviation the first time it is mentioned in the text, then you must use the abbreviation not the full name of the variable.
- Comments on the introduction:
In the introduction you use the abbreviations correctly, but there is a mistake with Socio-economic status. As stated in the abstract, SES refers to socio-economic status, in line 92, you have used SES as the abbreviation of socioeconomic. Check and fix this mistake.
Line 117 and 125, check if ACtive must have the two first words capitalized.
Line 119, “wasmanaged”, one space is missing.
Why don’t you abbreviate cardio-fitness performance?
- Comments on material and methods
Line 219, it is not necessary to write again that the SLJ is measured in cm.
With a sample size of 370 subjects, the appropriate test for normality is the Kolmogórov-Smirnov test.
- Comments on results:
Line 27, Table 1, should not be in bold.
Which test is used in table 1 to give the p-value?
Line 289, again, Table 2, should not be in bold.
- Comments on conclusions:
Check abbreviation of socioeconomic (SES), line 573.
- General comments:
This is the objective of the study: The present study aims to investigate several determinants of HRQoL, focusing specifically on SES and healthy lifestyle (PA and cardio-fitness performance) factors among Italian middle school students participating in the ACtive Breaks intervention.
When I read the conclusions, I am not able to see clearly whether the objective has been achieved and what clear implications the results of this work have.
Check all abbreviations across the document.
Some mistake mentioned above should be fixed
Author Response
Open Review
(x) I would not like to sign my review report
( ) I would like to sign my review report
Quality of English Language
( ) I am not qualified to assess the quality of English in this paper
( ) English very difficult to understand/incomprehensible
( ) Extensive editing of English language required
( ) Moderate editing of English language required
(x) Minor editing of English language required
( ) English language fine. No issues detected
Yes Can be improved Must be improved Not applicable
Does the introduction provide sufficient background and include all relevant references?
(x) ( ) ( ) ( )
Are all the cited references relevant to the research?
(x) ( ) ( ) ( )
Is the research design appropriate?
( ) (x) ( ) ( )
Are the methods adequately described?
( ) (x) ( ) ( )
Are the results clearly presented?
( ) (x) ( ) ( )
Are the conclusions supported by the results?
( ) ( ) (x) ( )
Comments and Suggestions for Authors
First of all, congratulations for the work done, then I will mention a number of changes and recommendations in order to obtain clearer and more accurate information.
Response: We would like to thank the reviewer for comments and suggestions that allow us to improve our work.
- Comments on the abstract:
Avoid the use of abbreviations in the abstract, and the if you use them, define the abbreviation the first time it is mentioned in the text, then you must use the abbreviation not the full name of the variable.
Response: Following the reviewer suggestion, we have revised the manuscript and completely removed all abbreviations from the abstract. Moreover, in the main body of the manuscript, any abbreviation used has been properly defined at its first mention as per standard academic writing conventions. From that point onwards in the text, we have consistently used the abbreviation, ensuring that the flow of the content remains unbroken and clear.
- Comments on the introduction:
In the introduction you use the abbreviations correctly, but there is a mistake with Socio-economic status. As stated in the abstract, SES refers to socio-economic status, in line 92, you have used SES as the abbreviation of socioeconomic. Check and fix this mistake.
Response: We apologize for the mistake. We have now fixed the sentence.
Line 117 and 125, check if ACtive must have the two first words capitalized.
Response: We apologize for the mistake. We fixed it.
Line 119, “wasmanaged”, one space is missing.
Response: Fixed.
Why don’t you abbreviate cardio-fitness performance?
Response: Following the reviewer’s recommendation, we have amended the manuscript to introduce and use the abbreviation "CF" (Cardio-Fitness). This change was implemented to streamline the readability of our text and maintain coherence with other abbreviations used in the manuscript.
- Comments on material and methods
Line 219, it is not necessary to write again that the SLJ is measured in cm.
Response: We revised it.
With a sample size of 370 subjects, the appropriate test for normality is the Kolmogórov-Smirnov test.
Response: We thank the reviewer for the thoughtful comment about our chosen test for normality. Given the sample size of 370 subjects, the Kolmogorov-Smirnov test would indeed be more appropriate for assessing the normality of our data. We appreciate the reviewer’s expertise and attention to detail in pointing out this methodological improvement. We have now reanalyzed our data using the Kolmogorov-Smirnov test for normality. The manuscript has been updated to reflect this change in methodology
- Comments on results:
Line 27, Table 1, should not be in bold.
Response: We revised it.
Which test is used in table 1 to give the p-value?
Response: We revised it.
Line 289, again, Table 2, should not be in bold.
Response: We apologize for the mistake, we revised it.
- Comments on conclusions:
Check abbreviation of socioeconomic (SES), line 573.
Response: We have now fixed the sentence.
- General comments:
This is the objective of the study: The present study aims to investigate several determinants of HRQoL, focusing specifically on SES and healthy lifestyle (PA and cardio-fitness performance) factors among Italian middle school students participating in the ACtive Breaks intervention.
When I read the conclusions, I am not able to see clearly whether the objective has been achieved and what clear implications the results of this work have.
Response: We would like to thank the reviewer for the suggestion. We revised the conclusion.
Check all abbreviations across the document.
Response: We checked.
Comments on the Quality of English Language: Some mistake mentioned above should be fixed
Response: We revised the text to comport with the reviewer's suggestion.
Submission Date
11 June 2023
Date of this review
13 Jun 2023 12:23:16

Reviewer 2 Report
Thank you for submitting the manuscript entitled “Association of Socioeconomic Factors and Physical Activity with Health-Related Quality of Life in Italian Middle School Children: An Exploratory Cross-Sectional Study”. Please see my comments below.
Abstract:
· Identifying associating factors is crucial but why only socioeconomic factors and physical activity. The introduction is not clear.
· The title and the aims are not consistent.
· What kind of Physical activity?
· “younger students and females reported higher HRQoL (β=- 0.139, P<.05, 95% CI: -0.254 – -0.023 and β=0.142, P><.05, 95% CI: 0.011 – 0.273 respectively). ><0.5….” what are the “p”?
Introduction
The introduction was written with unclear information. What exactly are the components of HRQOL? The paragraph to describe WHO’s HRQOL that lost the focus. It is difficult to understand the main point.
“From childhood onward, HRQoL takes on special meaning.” Is not clear. What are you going to mention? How do you understand the “subjective” HRQOL in children? What have been known in the past studies about HRQOL in children and adolescents?
The “PA” is a specific component? What is the PA in the past studies?
Material and methods
· The BRAVE study? Was your study a secondary analysis of the Brave study?
· What kinds of health issues or physical disabilities to be excluded?
· What about the setting? What about the sample size calculation?
Instrument
· There are numerous items for the HRQOL using KIDSCREEN-27. What types of the scoring of the domains? What about each reliability results?
· What about the scorings and reliability result of the other two instruments?
What about the study procedure? Was Ethics approval obtained before investigation? Who collected the data?
Results:
· How was the age range (10.89 to 15.34) obtained?
· Quite a large data were missing. The authors should explain. The analysis must be mentioned how to treat the missing data.
· P<0.05should be replaced by exact results.
Discussion
This part is too long. The authors should organize, summarize and interpret the results although various analytic methods were used. Based on the results, the authors should mention what results impact the current practice.
Other
professional English editing is needed.
professional editing is needed.
Author Response
Open Review
(x) I would not like to sign my review report
( ) I would like to sign my review report
Quality of English Language
( ) I am not qualified to assess the quality of English in this paper
( ) English very difficult to understand/incomprehensible
( ) Extensive editing of English language required
(x) Moderate editing of English language required
( ) Minor editing of English language required
( ) English language fine. No issues detected
Yes Can be improved Must be improved Not applicable
Does the introduction provide sufficient background and include all relevant references?
( ) ( ) (x) ( )
Are all the cited references relevant to the research?
( ) (x) ( ) ( )
Is the research design appropriate?
( ) ( ) (x) ( )
Are the methods adequately described?
( ) ( ) (x) ( )
Are the results clearly presented?
( ) ( ) (x) ( )
Are the conclusions supported by the results?
( ) ( ) (x) ( )
Comments and Suggestions for Authors
Thank you for submitting the manuscript entitled “Association of Socioeconomic Factors and Physical Activity with Health-Related Quality of Life in Italian Middle School Children: An Exploratory Cross-Sectional Study”. Please see my comments below.
Abstract:
- Identifying associating factors is crucial but why only socioeconomic factors and physical activity. The introduction is not clear.
Response: We revised the abstract according to the suggestion.
- The title and the aims are not consistent.
Response: We revised the aim to be more consistent with the title
- What kind of Physical activity?
Response: We revised the abstract to be more clear.
- “younger students and females reported higher HRQoL (β=- 0.139, P<.05, 95% CI: -0.254 – -0.023 and β=0.142, P><.05, 95% CI: 0.011 – 0.273 respectively). ><0.5….” what are the “p”?
Response: We apologize for the mistake, we fixed the sentences
Introduction
The introduction was written with unclear information. What exactly are the components of HRQOL? The paragraph to describe WHO’s HRQOL that lost the focus. It is difficult to understand the main point.
Response: We revised the introduction to be more clear.
“From childhood onward, HRQoL takes on special meaning.” Is not clear. What are you going to mention? How do you understand the “subjective” HRQOL in children? What have been known in the past studies about HRQOL in children and adolescents?
Response: We thank the reviewer for the thoughtful inquiry into the statement, "From childhood onward, HRQoL takes on special meaning." We apologize for any confusion.
The subjective nature of HRQoL in children, like adults, is drawn from the fact that it's the individual's perception of their well-being, encompassing not just physical health, but also emotional, social, and cognitive functioning. Given its subjective nature, HRQoL can differ greatly among children of similar health status based on their coping skills, resilience, support systems, and overall outlook.
In the context of childhood, HRQoL is particularly poignant. As children grow and progress through adolescence, they experience rapid changes in their physical, cognitive, emotional, and social capabilities. These significant maturational shifts, coupled with influential socialization factors, shape their well-being and lay the foundation for adjustment in later life. Therefore, an evaluation of HRQoL during this critical period offers profound insights into a child's psychosocial functioning and overall well-being.
Previous research into HRQoL in children and adolescents has indeed produced diverse findings. Much of this work has been disease-specific, focusing on conditions like diabetes, asthma, cancer, obesity, learning disabilities, and mental health conditions. These studies have greatly contributed to our understanding of the impact of such conditions on children's HRQoL. There has also been a concerted effort to develop measures for HRQoL in the context of epidemiological and public health research, with the aim of generating population-based data to inform policy and decision-making.
We understand your concerns about the clarity of this part of the text. To enhance readability, we had further refined this section in the manuscript to more explicitly outline the importance and subjective nature of HRQoL in children, drawing from past studies and our own work.
The “PA” is a specific component? What is the PA in the past studies?
Response: We apologize for the unclear definition of physical activity (PA). We revised the new paragraph “1.3 HRQoL and Physical Activity” to explain the definition of physical activity.
Material and methods
- The BRAVE study? Was your study a secondary analysis of the Brave study?
Response: Our research is not a secondary analysis of the BRAVE study, but rather an independent primary analysis conducted within the broader BRAVE project framework. Our exploratory cross-sectional study was conducted using data collected within the BRAVE study, with the aim of investigating the association of socioeconomic factors and physical activity with health-related quality of life in middle school children. The data for our study was collected and analyzed for the first time, which classifies our work as a primary analysis.
- What kinds of health issues or physical disabilities to be excluded?
Response: We thank the reviewer for pointing out the issue regarding the health issues or physical disabilities that led to the exclusion of participants from our study. In our study, we aimed to investigate the factors affecting physical activity and its relationship with the Health-Related Quality of Life in secondary school children. Therefore, our inclusion criteria were designed to ensure that the study population could actively participate in physical activities. In this context, our exclusion criteria involved any health issues or physical disabilities that might significantly interfere with or impact a student's physical activity performance. These include, significant cardiovascular, respiratory, or musculoskeletal disorders, neurological conditions, and severe visual or hearing impairments. The aim was to ensure the safety of the participants and to avoid potential confounding effects on the results due to conditions that significantly limit physical activity. We have now edited the methods section to include this information.
- What about the setting? What about the sample size calculation?
Response: We thank the reviewer for this comment. The study took place at a secondary school in Valsamoggia, Bologna, within the Emilia Romagna region of northeastern Italy.
Regarding the sample size calculation, this exploratory study did not undertake a formal sample size calculation prior to data collection. We used path analysis, which may result in higher Type II error rates with decreasing sample size. Nonetheless, the number of students in our study (224 students) exceeds the lower limit tested in simulation studies, thereby justifying the sample size used for our analysis. We acknowledge the need for a larger sample size to fully test the path model and ensure precision in the parameter estimates, including obtaining accurate standard errors. Despite these limitations, we believe the current study provides valuable insights into various socioeconomic, demographic, cardio-fitness, and physical activity measures that contribute to health-related quality of life in children.
Instrument
- There are numerous items for the HRQOL using KIDSCREEN-27. What types of the scoring of the domains? What about each reliability results?
Response: We have now added this information in the methods section.
Regarding the reliability of the KIDSCREEN-27, while we have not explicitly reported the Confirmatory Factor Analysis (CFA) results in the manuscript due to concerns about redundancy, we have performed it. In alignment with existing literature, the KIDSCREEN-27 tool has been extensively validated and shown to have excellent reliability across diverse populations and settings. However, we recognize the importance of detailing the reliability of our instruments, and therefore, we have revised our methods section to include a reference to the extensive validation and reliability testing of the KIDSCREEN-27 tool.
- What about the scorings and reliability result of the other two instruments?
Response: We would like to apologize for this lack of information. We revised the methods section adding the scoring system and the reliability of the instruments.
What about the study procedure? Was Ethics approval obtained before investigation? Who collected the data?
Response: As detailed in the methods section of our manuscript, this study was conducted under the approval of the Bioethics Committee of the University of Bologna. The "BRAVE" project received the committee's endorsement on 18 March 2022 (Protocol n. 63053), ensuring the ethical integrity of our research process. For data collection, physical education teachers were responsible for gathering cardio-fitness performance data from the study participants. Additionally, the questionnaire that was used to collect other relevant data was administered by the teachers of each class. These details ensure that the data collection process was conducted by professionals who interact with the students on a regular basis and are familiar with the research procedures. We have made sure to elucidate these points in the manuscript for transparency and accuracy.
Results:
- How was the age range (10.89 to 15.34) obtained?
Response: The age range of 10.89 to 15.34 years was obtained by calculating the difference between the day the survey was administered and each participant's date of birth. This calculation provided us with the precise age of each participant at the time of the survey. Subsequently, the minimum and maximum ages were identified, forming the range mentioned. We understand that this detail is crucial for the comprehension of our study's participant demographics, and we appreciate the inquiry for clarity. We have revised the manuscript to include this explanation about the age calculation to avoid any ambiguity.
- Quite a large data were missing. The authors should explain. The analysis must be mentioned how to treat the missing data.
Response: We appreciate the reviewer’s concern regarding the missing data in our study and the necessity for clarity on how it was addressed in our analysis. We agree that handling missing data transparently is crucial for maintaining the integrity of our findings. The missing data occurred due to incomplete responses or lack of certain measurements for some participants. To maintain the robustness of our analysis, we utilized listwise deletion, which involves excluding any participant with missing data on the variables included in our models. This approach was deemed appropriate as it ensures that our analyses are based on complete data, enhancing the validity of our results. Additionally, we wish to highlight that the "ACtive Breaks in Secondary School Children: The BRAVE Study" is an ongoing project. We plan to retrieve the missing data in subsequent waves of data collection, which will allow us to refine and expand our analyses in the future. We have revised the manuscript to explicitly state our method for handling missing data and our plans for subsequent data collection. We trust that this clarification addresses the reviewer’s query and ensures the transparency of our research methodology.
- P<0.05should be replaced by exact results.
Response: We fixed the manuscript.
Discussion
This part is too long. The authors should organize, summarize and interpret the results although various analytic methods were used. Based on the results, the authors should mention what results impact the current practice.
Response: We would like to thank the reviewer for this comment. We revised the discussion to comport with suggestions.
Other
professional English editing is needed.
Response: We revised the manuscript with a native english speaker.
Comments on the Quality of English Language: professional editing is needed.
Submission Date
11 June 2023
Date of this review
17 Jun 2023 19:49:05

Reviewer 3 Report
Thank you for giving me the time to review your manuscript. This manuscript is interesting for considering the association of socioeconomic factors and physical activity with health-related quality of life. Regarding the contents, the following revision should be considered.
The abstract
-The abstract should include specific research design descriptions.
Introduction
Generally, there is no paragraph writing. The background contains many paragraphs. The author should focus on theory building, the problems, and research question paragraphs. The first and second paragraphs should include general information on the quality of life and associated factors related to medical and social factors. Moreover, the third and fourth paragraphs should introduce the research question as the theoretical and conceptual framework, including socioeconomic factors the authors focused on in international contexts and research questions.
-This research lacks the evidence gap and research questions.
-This study focuses on one hospital. However, there is a lack of reasons why this research focuses on the hospital.
-The introduction should include this study's international contexts and research questions.
Method
-Why did this study focus on participants in one shool?
-Each variable included in this study should be explicitly explained.
-Sample calculation should be described clearly.
Discussion
With the same as the background, the authors should use paragraph writing for logical theory building.
This study should describe the limitation of sampling bias, the results' applicability to other settings, and the future investigation in the limitation part.
In the conclusion or discussion, the study’s strengths should be focused on international readers.
Conclusion
The conclusion section describes the result again. The authors should add a description of news regarding this research field.
Author Response
Open Review
( ) I would not like to sign my review report
(x) I would like to sign my review report
Quality of English Language
(x) I am not qualified to assess the quality of English in this paper
( ) English very difficult to understand/incomprehensible
( ) Extensive editing of English language required
( ) Moderate editing of English language required
( ) Minor editing of English language required
( ) English language fine. No issues detected
Yes Can be improved Must be improved Not applicable
Does the introduction provide sufficient background and include all relevant references?
( ) ( ) (x) ( )
Are all the cited references relevant to the research?
( ) ( ) (x) ( )
Is the research design appropriate?
( ) ( ) (x) ( )
Are the methods adequately described?
( ) ( ) (x) ( )
Are the results clearly presented?
( ) ( ) (x) ( )
Are the conclusions supported by the results?
( ) ( ) (x) ( )
Comments and Suggestions for Authors
Thank you for giving me the time to review your manuscript. This manuscript is interesting for considering the association of socioeconomic factors and physical activity with health-related quality of life. Regarding the contents, the following revision should be considered.
The abstract
-The abstract should include specific research design descriptions.
Response: As mentioned in the abstract, this study is cross-sectional.
Introduction
Generally, there is no paragraph writing. The background contains many paragraphs. The author should focus on theory building, the problems, and research question paragraphs. The first and second paragraphs should include general information on the quality of life and associated factors related to medical and social factors. Moreover, the third and fourth paragraphs should introduce the research question as the theoretical and conceptual framework, including socioeconomic factors the authors focused on in international contexts and research questions. Laura
Response: We would like to thank the reviewer for this comment. We revised the introduction as suggested.
-This research lacks the evidence gap and research questions.
Response: To the best of our knowledge this is the first study to explore the combined effects of SES and PA on HRQoL in Italian middle school students. As such, the study is poised to investigate this underrepresented subpopulation and fill this knowledge gap. We think that the changes in the introduction section emphasize this point.
-This study focuses on one hospital. However, there is a lack of reasons why this research focuses on the hospital.
Response: As mentioned in the methods sections, this study focused on children attending single secondary school, not a hospital.
-The introduction should include this study's international contexts and research questions.
Response: We have enriched the introduction by including more international references to underline the global relevance of our findings. We have also dedicated a separate subsection to comprehensively discuss the research questions. This section will help to provide more clarity about the direction of our research and its importance within the context of international studies on the subject.
Method
-Why did this study focus on participants in one shool?
Response: The BRAVE study was performed in a single secondary school of Bologna which was interested in participating in the project.
-Each variable included in this study should be explicitly explained.
Response: We explained in detail all the measurement performed in the methods section creating different paragraph :
2.3 Main outcome measure: health-related quality of life
2.4 Physical Activity Questionnaire for Older Children
2.5 Physical fitness measurements
2.6. Socioeconomic variables
2.7. Covariates
We would ask the reviewer which specific measure is missing.
-Sample calculation should be described clearly.
Response: Regarding the sample size calculation, this exploratory study did not undertake a formal sample size calculation prior to data collection. We used path analysis, which may result in higher Type II error rates with decreasing sample size. Nonetheless, the number of students in our study (224 students) exceeds the lower limit tested in simulation studies, thereby justifying the sample size used for our analysis. We acknowledge the need for a larger sample size to fully test the path model and ensure precision in the parameter estimates, including obtaining accurate standard errors. Despite these limitations, we believe the current study provides valuable insights into various socioeconomic, demographic, cardio-fitness, and physical activity measures that contribute to health-related quality of life in children.
Discussion
With the same as the background, the authors should use paragraph writing for logical theory building.
Response: We revised the discussion to be consistent with the introduction section.
This study should describe the limitation of sampling bias, the results' applicability to other settings, and the future investigation in the limitation part.
Response:
In the conclusion or discussion, the study’s strengths should be focused on international readers.
Response: We would like to thank the reviewer. We revised the conclusion to reinforce this concept.
Conclusion
The conclusion section describes the result again. The authors should add a description of news regarding this research field.
Response: Response: We would like to thank the reviewer for this comment. We believe that much information was included in the conclusion section in particular regarding future implications. However, in order to comport with the reviewer's suggestion we revised the conclusion.
Submission Date
11 June 2023
Date of this review
17 Jun 2023 08:35:40

Reviewer 4 Report
Review:
Line 40-42: “…..as well as non-medical aspects of functioning including emotional, 40 social, and cognitive functioning [1]. In general, most medical and health professionals 41 consider HRQoL as…”
Why do you continually appoint doctors and medical and clinical health professionals, they should unify it so as not to be redundant
Line 55-57: “From a measure-54 ment standpoint, HRQoL assesses important aspects of health that are subjective and not 55 detected by traditional physiological and clinical assessments”
As which?
Line 60-62: Although much of the early work using the WHO perspective emphasized health in relationship to functional status and disability, both conceptual models of HRQoL and assessment strategies now incorporate both individual and environmental factors that in-fluence well-being [7]
Perhaps the WHO quote should be put and name the main individual factors to which they refer.
Line 152: Main outcome measure: health-related quality of life This section could have subsections for better visualization or a table
Line 227-228: What instrument was the child weighed with? Was it around the same time for everyone? How much was removed when weighing the child, (to adjust the weight of the clothes and shoes, or was he weighed barefoot? 1kg, 05kg?
Conclusion section: The conclusions section lacks specific conclusions that help determine which socioeconomic and physical exercise factors are the most influential in a clear and concise manner.
The study has a lot of statistical development that is not reflected in conclusions, which are too general
Author Response
Open Review
( ) I would not like to sign my review report
(x) I would like to sign my review report
Quality of English Language
(x) I am not qualified to assess the quality of English in this paper
( ) English very difficult to understand/incomprehensible
( ) Extensive editing of English language required
( ) Moderate editing of English language required
( ) Minor editing of English language required
( ) English language fine. No issues detected
Yes Can be improved Must be improved Not applicable
Does the introduction provide sufficient background and include all relevant references?
(x) ( ) ( ) ( )
Are all the cited references relevant to the research?
( ) (x) ( ) ( )
Is the research design appropriate?
( ) (x) ( ) ( )
Are the methods adequately described?
( ) (x) ( ) ( )
Are the results clearly presented?
(x) ( ) ( ) ( )
Are the conclusions supported by the results?
( ) ( ) (x) ( )
Comments and Suggestions for Authors
Review:
Line 40-42: “…..as well as non-medical aspects of functioning including emotional, 40 social, and cognitive functioning [1]. In general, most medical and health professionals 41 consider HRQoL as…”
Why do you continually appoint doctors and medical and clinical health professionals, they should unify it so as not to be redundant
Response: We have revised the manuscript to eliminate this redundancy. We now use a unified term that encompasses both categories - "health professionals". We believe that this term effectively conveys the group we're referring to without being overly repetitive.
Line 55-57: “From a measure-54 ment standpoint, HRQoL assesses important aspects of health that are subjective and not 55 detected by traditional physiological and clinical assessments”
As which?
Response: We thank the reviewer for the valuable comment. In response to the query about what subjective aspects of health Health-Related Quality of Life (HRQoL) assesses that are not detected by traditional physiological and clinical assessments, we provide the following clarification:
HRQoL incorporates dimensions such as mental health, emotional wellbeing, social functioning, role limitations due to physical or emotional health problems, and general health perceptions. These aspects capture the individual's perceived impact of health status on quality of life, which are inherently subjective and may not be fully captured by traditional physiological and clinical measures such as blood tests, imaging studies, or physical examination findings.
We have updated our manuscript to explicitly mention these subjective dimensions assessed by HRQoL. The revised sentence reads: "From a measurement standpoint, HRQoL assesses important aspects of health that are subjective and not detected by traditional physiological and clinical assessments, such as mental health, emotional wellbeing, social functioning, role limitations due to physical or emotional health problems, and general health perceptions."
Line 60-62: Although much of the early work using the WHO perspective emphasized health in relationship to functional status and disability, both conceptual models of HRQoL and assessment strategies now incorporate both individual and environmental factors that in-fluence well-being [7]
Perhaps the WHO quote should be put and name the main individual factors to which they refer.
Response: We have elaborated on the individual factors that the WHO perspective emphasizes in relation to HRQoL. These factors primarily include physical health, mental health, level of independence, social relationships, personal beliefs, and their interaction with environmental factors. We have incorporated the WHO's definition of HRQoL into our manuscript, and outlined these individual factors. The revised section now reads: "Although much of the early work using the WHO perspective emphasized health in relation to functional status and disability, both conceptual models of HRQoL and assessment strategies now incorporate both individual and environmental factors that influence well-being [7]. According to the WHO, HRQoL is a broad-ranging concept that encompasses the individual's physical health, psychological state, level of independence, social relationships, personal beliefs and their relationship to salient features of their environment."
Line 152: Main outcome measure: health-related quality of life. This section could have subsections for better visualization or a table
Response: We are very grateful for the suggestion. However, we believe that referring exclusively to KIDSCREEN-27, a single questionnaire, it is useful to maintain a single section and not divide it into further subsections. Inserting a table would risk making the article heavier, as there are 27 items in the questionnaire and the article is already full of tables in the results section.
Line 227-228: What instrument was the child weighed with? Was it around the same time for everyone? How much was removed when weighing the child, (to adjust the weight of the clothes and shoes, or was he weighed barefoot? 1kg, 05kg?
Response: We utilized electronic scales that were calibrated before each weighing session to ensure the most accurate readings. To standardize the measurement procedure, all weigh-ins were conducted in the morning to minimize variations that can occur throughout the day. Children were asked to remove heavy clothing items, shoes, and any items in their pockets before stepping onto the scale. They were weighed barefoot. No specific weight was subtracted from the raw readings as the weight of remaining clothing items was assumed to be minimal and consistent across participants.
Conclusion section: The conclusions section lacks specific conclusions that help determine which socioeconomic and physical exercise factors are the most influential in a clear and concise manner.
Response: We thank the reviewer for the insightful comment regarding the clarity and specificity of our conclusions. In response to the suggestion, we have revisited our conclusion section and refined our statements to more explicitly identify the most influential socioeconomic and physical exercise factors. We trust that this revision brings more clarity and precision to our conclusions and addresses your comments effectively.
The study has a lot of statistical development that is not reflected in conclusions, which are too general
Response: We understand your concern about the gap between the comprehensive statistical analyses we've carried out and the relatively general conclusions drawn from them.
In response to the reviewer’s feedback, we have revised our conclusion section to better reflect the results of our statistical analyses. We now specify the most influential socioeconomic and physical activity factors on health-related quality of life, drawing direct links to our statistical findings.
The revised conclusions section reads as follows: "Our study, utilizing rigorous statistical methods, demonstrates a multifaceted relationship between PA, demographic factors, CF measures, and HRQoL in middle school children. Our data reveal a pronounced impact of regular PA, socio-economic factors, and CF on a child's HRQoL, highlighting the necessity of these factors for improving the overall well-being of children. The findings suggest a distinct health gradient linked to the education and employment status of parents, with children of better educated and employed parents showing a higher HRQoL. This understanding may serve as a foundation for creating interventions targeted at improving socio-economic disparities to achieve more equitable health outcomes for children across various demographic backgrounds.
Furthermore, our study encourages the inclusion of a broader range of measures in future research, such as cultural influences, nutrition, and family characteristics. We believe that longitudinal studies investigating causal relationships between these aspects, along with household income data, will provide a more comprehensive understanding of the factors influencing children's well-being. Additionally, this study underscores the need for future research to evaluate the impact of targeted PA interventions and explore potential mediators and moderators of these relationships. The insights from such studies could be pivotal for stakeholders in education, public health, and health policy sectors to formulate strategies that promote access to, and engagement in, PA, thereby fostering healthful choices in middle school children.
Our findings, thus, are a clarion call for a comprehensive, multifactorial approach in fostering a healthier, more resilient generation of youth, better equipped to navigate the challenges of adolescence and adulthood."
Submission Date
11 June 2023
Date of this review
15 Jun 2023 13:48:28

Round 2
Reviewer 2 Report
Thanks for the revised manuscript. This version has been improved a lot. I have minor comments to improve the manuscript for your consideration.
1. Abstract:
- PA should be mentioned in full term at the first time.
2. Introduction:
-It's NOT necessary to give a subheading e.g., "Health-related quality of life". It is important to follow the flow to describe the phenomena/contrast in the manuscript.
3. Results:
-p=n.s, the n.s. should be replaced with the exact number.
4. Discussion:
-It is NOT necessary to mention the subheading, e.g., "main findings" but important to follow the flow to interpret your findings one by one.
Author Response
Quality of English Language
( ) I am not qualified to assess the quality of English in this paper
( ) English very difficult to understand/incomprehensible
( ) Extensive editing of English language required
( ) Moderate editing of English language required
( ) Minor editing of English language required
(x) English language fine. No issues detected
Yes Can be improved Must be improved Not applicable
Does the introduction provide sufficient background and include all relevant references?
( ) (x) ( ) ( )
Are all the cited references relevant to the research?
(x) ( ) ( ) ( )
Is the research design appropriate?
(x) ( ) ( ) ( )
Are the methods adequately described?
(x) ( ) ( ) ( )
Are the results clearly presented?
( ) (x) ( ) ( )
Are the conclusions supported by the results?
( ) (x) ( ) ( )
Comments and Suggestions for Authors
Thanks for the revised manuscript. This version has been improved a lot. I have minor comments to improve the manuscript for your consideration.
- Abstract:
- PA should be mentioned in full term at the first time.
Authors: We apologize for the mistake. It has now been corrected.
- Introduction:
-It's NOT necessary to give a subheading e.g., "Health-related quality of life". It is important to follow the flow to describe the phenomena/contrast in the manuscript.
Authors: We corrected the introduction as suggested.
- Results:
-p=n.s, the n.s. should be replaced with the exact number.
Authors: We apologize for the inconsistency. We have now provided the exact values.
- Discussion:
-It is NOT necessary to mention the subheading, e.g., "main findings" but important to follow the flow to interpret your findings one by one.
Authors: We have now removed the subheadings as suggested.
Submission Date
11 June 2023
Date of this review
13 Jul 2023 05:20:12

Reviewer 3 Report
The authors improved the contents for the publication.
Author Response
Quality of English Language
(x) I am not qualified to assess the quality of English in this paper
( ) English very difficult to understand/incomprehensible
( ) Extensive editing of English language required
( ) Moderate editing of English language required
( ) Minor editing of English language required
( ) English language fine. No issues detected
|
|
Yes |
Can be improved |
Must be improved |
Not applicable |
|
Does the introduction provide sufficient background and include all relevant references? |
(x) |
( ) |
( ) |
( ) |
|
Are all the cited references relevant to the research? |
(x) |
( ) |
( ) |
( ) |
|
Is the research design appropriate? |
(x) |
( ) |
( ) |
( ) |
|
Are the methods adequately described? |
(x) |
( ) |
( ) |
( ) |
|
Are the results clearly presented? |
(x) |
( ) |
( ) |
( ) |
|
Are the conclusions supported by the results? |
(x) |
( ) |
( ) |
( ) |
Comments and Suggestions for Authors
The authors improved the contents for the publication.
Authors: We thank the reviewer for the valuable feedback.
Submission Date
11 June 2023
Date of this review
12 Jul 2023 22:11:06
Quality of English Language
(x) I am not qualified to assess the quality of English in this paper
( ) English very difficult to understand/incomprehensible
( ) Extensive editing of English language required
( ) Moderate editing of English language required
( ) Minor editing of English language required
( ) English language fine. No issues detected
|
|
Yes |
Can be improved |
Must be improved |
Not applicable |
|
Does the introduction provide sufficient background and include all relevant references? |
(x) |
( ) |
( ) |
( ) |
|
Are all the cited references relevant to the research? |
(x) |
( ) |
( ) |
( ) |
|
Is the research design appropriate? |
(x) |
( ) |
( ) |
( ) |
|
Are the methods adequately described? |
(x) |
( ) |
( ) |
( ) |
|
Are the results clearly presented? |
(x) |
( ) |
( ) |
( ) |
|
Are the conclusions supported by the results? |
(x) |
( ) |
( ) |
( ) |
Comments and Suggestions for Authors
The authors improved the contents for the publication.
Authors: We thank the reviewer for the valuable feedback.
Submission Date
11 June 2023
Date of this review
12 Jul 2023 22:11:06
